# Verification of Usability of Medical Image Data Using Projective Photography for Designing Clothing for Breast Cancer Patients

**Youn Joo Kim** 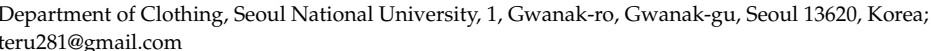

Department of Clothing, Seoul National University, 1, Gwanak-ro, Gwanak-gu, Seoul 13620, Korea; teru281@gmail.com

**Abstract:** Manufacturing a customized mastectomy bra, using medical images obtained for breast cancer treatment, could be suggested as an alternative instead of the anthropometric method. However, the breast shape of a medical image is deformed from the anthropometric method as the measurement posture is different between the anthropometric method for making clothes and the medical image. As a breast consists of adipose tissues and a few muscles without bones, there is a possibility that a bra can be manufactured if the volume is constant. Therefore, a hypothesis was established that the volume of the breast would be constant, even if the measurement methods were different. As a result of the comparison of 3D-SIM and PPM by MRI, 18 items could be measured simultaneously. Nine items showed differences according to the measurement method. The next step in the case of 3D-SIM was calculating the volume by separating the breast shape into a cone and a hemispherical shape; in the case of MRI, an ellipsoidal volume formula was applied. A t-test was performed on the results obtained, showing no significant difference. Therefore, it was proven that the volume of the breast does not change despite the difference in the measurement and the measurement method.

**Keywords:** mastectomy; bra; breast volume; MRI; anthropometry

## 1. Introduction

Mastectomy patients who have undergone breast tissue removal for breast cancer treatment experience not only physical pain, but also psychological anxiety and damage to the physical image, which may restrict them from social activities and affect their human relations [1–4]. According to the National Cancer Information Center [5], in 2018, 64.8% of patients with breast cancer in Korea underwent a total mastectomy, which is the total excision of skin and lymph nodes, including the entire breast tissue and nipples, and only 26% underwent breast-conserving surgery, which is the removal of cancer without breast tissue. Patients who underwent surgical excision experienced physical pain, such as an imbalance in body shape due to loss of breasts, muscular pain, paresthesia, lymphedema, and muscle weakness, while approximately 30% experienced functional obstruction [6]. Additionally, in a study by Al-Ghazal et al. [7], patients who had undergone mastectomy developed diverse physical symptoms, including scoliosis or shoulder pain due to an imbalance in their body weight. Another study revealed that 78% of mastectomy patients had their upper body distorted by up to 3° [8].

Although an increasing number of patients are undergoing breast reconstruction, surgery is not always an option owing to medical issues, which is why there remains a high percentage of breast cancer patients who do not have their breasts reconstructed. Therefore, mastectomy bras and breast prostheses are suggested as a substitute for breasts [9,10]. This suggestion is related to a study reporting that wearing a breast prosthesis improved upper limb motility [11]. Therefore, the mastectomy brassiere and breast prosthesis will help reduce physical pain along the mastectomy, and thus need suitable fitting. However, wearing inappropriate mastectomy bras and breast prostheses may instead cause physical deformation or musculoskeletal pain. Gho et al. [12] claimed that the unbalanced

movement of the left and right breasts causes patients to recognize the incompleteness of their body image, thereby causing psychological distress. Therefore, each bra should be personalized and customized for the right bra fit that reflects the body shapes of mastectomy patients. The need to solve the left–right imbalance and variability in the position of a brassiere for breast cancer patients has been mentioned in numerous studies [13,14], yet functionality has not been improved. This problem necessitates the need to study and produce bras that are suitable for the body shapes of mastectomy patients. At present, there are insufficient human body data on total mastectomy patients, which would have aided in the production process. Moreover, mastectomy patients have different areas or forms of excision, such as breasts and axillary lymph nodes, depending on the stage or method of surgery, which makes it difficult to categorize the body shapes for certain areas of measurement. In addition, the current human body measurement method used for manufacturing has a disadvantage: accurate measurement cannot be made because the bottom surface of the breast cannot be determined. In addition, as mastectomy patients tend to avoid revealing their affected parts [15,16], it may be a burden on them to expose their excised breasts to others to collect human body data for customizing bras. Thus, it is necessary to establish contactless body measurement methods to minimize repeated exposure of the body as well as skin contact to ensure the patients' mental stability.

Methods of collecting breast-related human body data include using human body images, such as magnetic resonance imaging (MRI) and computed tomography (CT) for medical purposes or anthropometric 3D body scanning for ergonomic purposes. As MRI is scanned from multiple angles, more stereoscopic human body data can be obtained. Thus, MRI is used for various purposes, such as restoration of the human body or volume measurements and for ergonomic purposes. Nonetheless, the shape of breasts may vary in MRI and 3D body scanning depending on the posture, which results in different measurements of detailed items. However, to the best of our knowledge, there has been no research on the posture that best displays the characteristics of breasts and is thus suitable for bra design. Therefore, this study uses MRI data, a medical imaging technique, to measure similar results of breast volume as 3D body scanning and compare it using anthropometric data, thereby identifying that the volume of the two data is the same. Nevertheless, as breasts are soft and easily change shape, cups in various designs can be worn as long as the under-bust circumference and bra cup size match [17].

Consequently, this study hypothesizes that the volume of breasts will remain the same, even when the shape changes because breasts are comprised of adipose substances and do not have bones. If the volume is constant, the drafting of the bra pattern will be possible. If so, it will be possible to make clothes, even if medical images are used, not necessarily according to the human body measurement method. Therefore, this study assumes that the breast volume is the same, despite the difference in measurement methods. To prove the possibility, this study was designed to illuminate areas showing differences in measured values for different measurement methods and to compare breast volumes. Therefore, this study will test the hypothesis that, despite the different correlations and measurement methods, the breast volume remains the same by measuring the volume of the two different visual data. If the hypothesis is proven, it is possible to suggest the possibility that human data collected for medical purposes can be used for manufacturing clothing. Therefore, it is possible to supplement the limitations of the 3D-SIM measurement method and use the measurement method considering the psychological state of the patient in the manufacture of ergonomic clothing. In addition, the possibility of convergence research in medical and clothes engineering is presented by introducing various methods of utilizing data that have been mainly used for medical purposes.

## 2. Contactless Human Body Data Measurement

### 2.1. Projective Photography Measurement (PPM)

Projective photography measurement (PPM) is a method that analyzes images, projecting the inside of the human body using radiation and magnetic force mostly for medical

purposes. It is also referred to as medical image analysis. PPM is relatively simple and has high reliability [18]. Diagnostic imaging mostly used to diagnose breast cancer includes ultrasonography, mammography, CT, positron emission tomography and computed tomography (PET-CT), and MRI [19,20]. The methods and characteristics of imaging for medical purposes are shown in Table 1. MRI and CT are the methods that can be used to obtain 3D images, and they can measure volume, unlike 2D imaging.

Ultrasonography cannot measure the outline of breasts because it scans only localized areas of breasts. It also presses the breasts with a sensor, which changes their shape; thus, the outline is not projected. Mammography is taken in the standing position, but the device presses the breasts from up to down or on both sides, changing the shape of the breasts, like ultrasonography. It only scans the areas where the rib cage and pectoral muscles are not revealed, and thus the precise outline of the breasts cannot be measured. CT and PET-CT create tomography images by scanning the transverse plane of the body and reconstructing them on a computer. It is easy to measure human body data with these methods, as there is a clear distinction between the skeletal structure and internal tissue. However, as they are scanned in a lying position, breasts comprised of adipose tissue cannot maintain their original shape. Furthermore, it is impossible to measure items related to the sagittal or coronal plane, as it only measures the transverse plane. Moreover, frequent CT imaging is also identified as a problem, owing to the risk of radiation exposure. However, MRI can obtain 3D data using a separate program because it collects data from the coronal, sagittal, and transverse plane, simultaneously, and thus it is used in various medical fields [21–23]. As such, four out of five PPMs cannot obtain accurate measurements from the outline of the body, owing to excessive breast shape deformation; by comparison, MRI can be scanned from multiple angles. Thus, obtaining 3D images of a human body or applying them to mammoplasty is common in the medical world, producing data with high reproducibility of the human body. MRI can also identify the locations of bones and muscles in clothing, as it scans not only the inside, but also the outline of the body, can obtain the volume of breasts and breast tissue [24], and has no risk of radiation exposure [25].

PPM data are required in the process of treatment and cancer tracking. Thus, if this data can be used in apparel production, it might not be necessary to take body measurements again for customization. Therefore, this study selected MRI data as the comparison target, as they have similar characteristics to three-dimensional surface imaging measurement (3D-SIM) data and have the highest human body reproducibility.

Medical image data are obtained multiple times in the treatment process; thus, various medical image data obtained during the treatment process can help analyze various changes in the human body before and after surgery. Thus, if there are pre- and post-surgery data, the same ratio can be applied to the weight and form of both breasts. Moreover, because data are obtained in the treatment process, body measurements do not have to be taken separately. Medical image data of mastectomy patients obtained during cancer treatment include ultrasonography, mammography, CT, and MRI data. MRI data bring the least change to the human body form and can easily measure the detailed items of breasts by scanning the body from multiple angles. Moreover, the 90° prone position is considered the most suitable method to separate the breasts from the rib cage in plastic surgery [26], and the breast separation in this posture is most suitable for forming the asymmetry of both breasts by measuring the breast volume. Furthermore, MRI data can be used to make a pattern of the human body or be applied to mammoplasty, which is common in the medical world, indicating that it has high reproducibility of the human body. Therefore, the human body data of MRI have high applicability to apparel production. Moreover, MRI scans the inside of the human body and the outline, thereby identifying the location of bones and muscles in clothing. Thus, MRI data are suitable for designing underwear, sportswear, and compression wear, that are affected by the skeletal structure, muscle location, and bust depth.

<div align="center">**Table 1.** Methods and characteristics of imaging for medical purposes.</div>

| Name | Purpose | Radiation Use | Precision | Characteristic | Measurement Time (min) | Posture |
|---|---|---|---|---|---|---|
| Ultrasonography | Primary examination of disease | None | Low | Can identify the type of tumor through radio-logical finding | 10–20 | Lying position |
| Mammography | Primary examination of disease | Ionizing radiation | Low | High contrast imaging that shows the difference in X-ray absorption between normal tissue and lesion | 5–10 | Standing position |
| CT | Stage identification Metastasis identification | Ionizing radiation | High | Tomography of the transverse plane of the body with multi-angle X-ray penetration | 10–15 | Lying position |
| PET-CT | Malignant cancer identification Stage identification/Metastasis identification | Ionizing radiation | High | Scanning the distribution within the body by injecting radio-pharmaceutical (F-FDG) that discharge positrons | 20–40 | Lying position |
| MRI | Stage identification Metastasis identification | None | High | Can check high-contrast and high-resolution coronal plane, sagittal plane, transverse plan | 10–15 | Prone position |

### 2.2. Three-Dimensional Surface Imaging Measurement (3D-SIM)

The three-dimensional surface imaging measurement is used to measure and analyze the surface of the human body using images from a 3D scanner. The anthropometric method is applied to 3D-SIM data, and it has been used mostly in ergonomics and apparel studies since the 1980s [27–29]. As it is not affected by breathing, compared to direct body measurements, the measures obtained can be relatively accurate. Moreover, the body is scanned in a short time—less than 20 s—without physical contact, and human body data to be measured can be quickly extracted using a computer program, which can be reused anytime. Furthermore, 3D-SIM can resolve the issues of reliability and accuracy of the measurement technique or method when directly measured by humans. Hertzberg et al. [30] compared measurement by humans with 3D-SIM data and discovered that 3D-SIM is much more accurate than the body measurement by humans. Loughry et al. [31] also argued that 3D-SIM is much more accurate, as it does not press or transform breast tissue while measuring breast volume [32,33]. Other studies also highly rated 3D-SIM images using a computer [34–36]. However, 3D-SIM can be used only in a place with expensive scanning equipment, and the patients must be partly naked during the measurements. Gowns are provided to facilitate body measurements, but soft tissue, such as breasts, can be deformed or pressed by the gowns. Surface pressure caused by the gowns may change the measurement data.

### 3. Material and Method

#### 3.1. Selection of Subjects

This study provides basic data for the manufacture of bras for breast cancer patients, and the age and breast shape are limited in the recruitment of subject. First, this study limited the age scope of mastectomy patients. The age group for breast resection for breast cancer is mainly 35–64 years old [37]. In addition, according to the statistical results, the range with the highest incidence rate of breast cancer patients was found to be in their 40s and 50s. Therefore, the age range of the participants was limited to women in their 40s–50s [37]. In addition, the range was limited to patients who had one breast removed for bra development. Therefore, the subjects had their inner breasts completely

excised. Mastectomy patients in their 40s and 50s who were undergoing breast cancer treatment were recruited at Seoul National University Hospital (IRB No. 1007/211-325). After checking and filtering the age, image data status, and scope of imaging in the MRI data, the data of 21 patients were collected. Thereafter, of the 21 patients, four volunteers were selected and their data were used in 3D-SIM analysis. The participants were marked as A, B, C, and D.

*3.2. Selection of Breast-Related Measurement Items*

For breast measurement, items (Tables 2–4 and Figure 1) that represent the size and shape of the upper body and are necessary for identifying the breast shape were selected based on previous studies [38–40]. The reference point and measurement items were based on the glossary for body measurements by Size Korea [41]. A total of 20 items were selected for breast volume measurements, such as circumference (2), depth (2), breadth (3), and detailed items of breasts (13) (Table 2). Furthermore, reviewing human body data related to the measurement items, a high reliability (Cronbach'$\alpha$) score of 0.853 was recorded.

The 3D-SIM and MRI human body data of the four participants were analyzed using descriptive statistics and a t-test with IBM SPSS Statistics for Windows Version 22.0 (IBM Corp, Armonk, NY, USA). The methods of measurement for the items are shown in Table 4.

**Table 2.** Measurement method of breast details.

| Measurement Items | Area |
|---|---|
| Circumference (2) | Bust circumference/Under-bust circumference |
| Depth (2) | Bust depth/Under-bust depth |
| Breadth (3) | Bust breadth/Under-bust breadth |
| Detailed items of breasts (13) | Superior bust surface length/Inferior bust surface length/Lateral bust surface length/Medial bust surface length/Superior bust straight length/Inferior bust straight length/Lateral bust straight length/Medial bust straight length/Mid-shoulder to bust point straight length/Verge surface/Upper bust slope angle/Lower bust slope angle/ Length between central line and bust point |

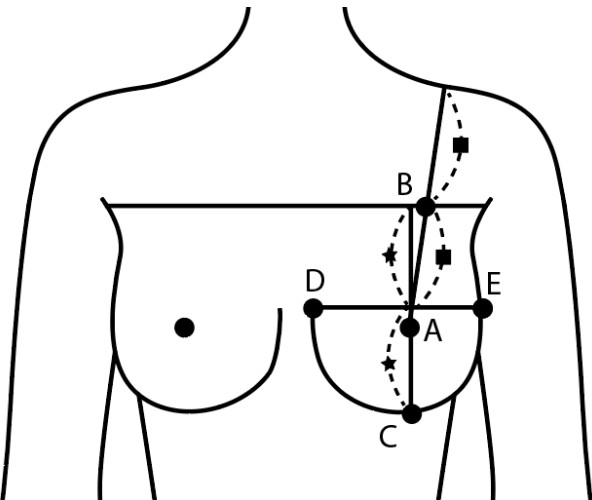

**Figure 1.** Measuring point on the bust (A: Bust Point, B: Superior Bust Point, C: Inferior Bust Point, D: Medial Bust Point, E: Lateral Bust Point).

**Table 3.** Measurement method of breast details.

| Point | Measurement Items | Area |
|---|---|---|
| A | Bust Point | The apex of the bust |
| B | Superior Bust Point | Intersection point horizontal line passing through the anterior axillary fold points and the crossing line from mid-shoulder points to the bust point. |
| C | Inferior Bust Point | Vertical drop line from the bust point. |
| D | Medial Bust Point | The crossing point of the verge line and inside point on the horizontal line through bisector point which is crossing the line from the inferior bust point. |
| E | Lateral Bust Point | The crossing point of verge line and outside point on the horizontal line through bisector point which is crossing the line from the inferior bust point |

**Table 4.** Measurement method for items.

| Measurement Items | Measurement Method |
|---|---|
| Bust surface circumference | Horizontal circumference measured along the surface line from the actual bust point level |
| Bust depth | Maximum depth including bust from the actual bust point level |
| Bust breadth | The widest horizontal distance from the plane at the actual bust point level |
| Under-bust surface circumference | Horizontal circumference from the Under-bust point |
| Under-bust depth | Maximum depth including bust from the Under-bust breadth |
| Bust breadth | The widest horizontal distance from the plane at the Under-bust point level |
| Upper bust slope angle | The angle of superior bust point, bust point, and line connecting a 90-degree line to bust point |
| Lower bust slope angle | The angle of bust point, Inferior bust point, and connecting a 90-degree line to bust point |
| Superior bust-length | Straight length from bust point to the superior bust point |
| Inferior bust-length | Straight length bust point to the inferior bust point |
| Lateral bust-length | Straight length from bust point to the lateral bust point |
| Medial bust-length | Straight length from bust point to the medial bust point |
| Superior bust surface length | Surface length from bust point to superior bust point |
| Inferior bust surface length | Surface length from bust point to inferior bust point |
| Lateral bust surface length | Surface length from bust point to the lateral bust point |
| Medial bust surface length | Surface length from bust point to the medial bust point |
| Chest depth | Length from a central point of the chest to bust point |
| bust point depth | Length from rib cage to bust point |
| Central line-bust point length | Length between anterior central line and bust point |
| Circumference surface length | Surface length of bust Medial bust point-bust point-Lateral bust point |

*3.3. MRI Image Obtaining Process*

The MRI data concerned breast cancer at Seoul National University Hospital, assent to the patient was used to analyze the breast-related measurement items (IRB No. 1007/211-325). The patients had to assume a position to take the MRI: their breasts hanging down the hole while lying prone on a bed. Usually, MRI data that scanned the upper body from the anterior neck point to waist circumference were used.

Data collected from the MRI scanner (Biograph MR; Siemens Healthcare, Erlangen, Germany) at Seoul National University Hospital were used, and the picture archiving and communication system (PACS) was used for the analysis and human body data measurement (Figure 2).

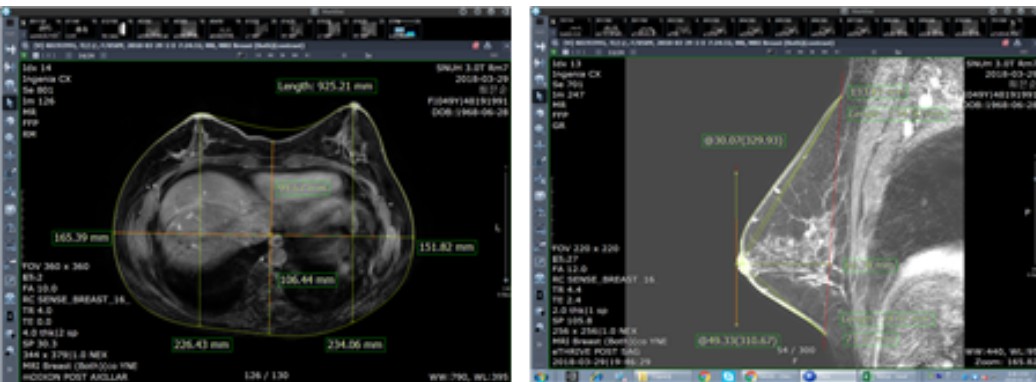

**Figure 2.** Measurement of detailed items of breasts using PACS.

*3.4. 3D-SIM Obtaining Process*

Data for the 3D-SIM were collected from four participants who volunteered to participate in this study (IRB No. 1808/003-001). The measurements were taken from 2 to 17 November 2018. The participants were to hold up their arms to 45°, 90°, and stretch as upward as they could, which are anthropometric postures, while a 360° upper body scan was conducted. The scanning of each motion took approximately 45 s. They were to inhale and remain still for as long as they could so that the rib cage was not expanded while scanning. There was a 10-min break between motions so that the participants could relax. The 10-min break was based on the speed of saving the 3D images, while also referring to previous studies on measuring bra stress [42–44] that designated the pause between protocols as 10 min. The process of 3D-SIM is shown in Figure 3.

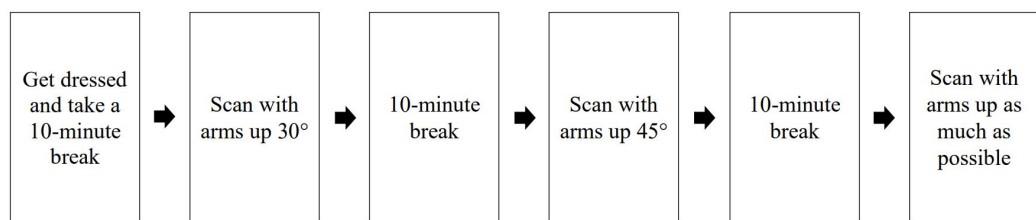

**Figure 3.** Process of 3D-SIM.

The 3D handheld scanner used in this study is the Artec EVA 3D scanner (Artec Group Company, Santa Clara, CA, USA) that can obtain precise images in high resolutions by quickly and accurately scanning the target based on safe and structured beam scanning technology. However, it requires a longer time compared with a full-body scanner, and it may also cause physical fatigue or burden on the researcher performing the scan. Nevertheless, the scope and angles of scanning are extensive, thereby minimizing data loss on excessively curved areas of the body. This scanner was used in this study, as it is optimized for scanning areas that frequently have missing values on full-body scanners, such as underarm and under-bust circumference lines.

If the researcher or participant rotates the scanner themselves for 3D-SIM scanning, the form may shake, which is why a 360-degree rotatable motorized rotating plate was used (Figure 4). A motorized rotating plate is designed to control the rotating speed and direction. However, participants may feel dizzy, owing to the rotating power of the plate in a standing position, and 3D images may overlap owing to lost postures, which was proven during the pilot experiment. In addition, considering that the participants are breast cancer patients, they were to remain seated on a light chair. A study claimed that the sitting position increases under-bust circumference [45,46], but the pilot experiment proved that the error range can be reduced to under 0.7 cm if a participant sits in a correct posture and remains tense, which is smaller than the error range of 2.096 cm in previous studies. This size is also included within the total error range of the product (±1 cm); therefore, the

sitting position can be used. Although the scanning is done in a sitting position, each scan takes 45 s and the body may shake or move along with the gaze. Thus, except for the space where the researcher performed the scan, a 3 cm-wide tape was attached to a board at eye level so that the participants could fix their gaze on the tape in rotation, thereby limiting image overlapping (Figure 4).

Artec Studio 10 Professional (Artec Group Company, Santa Clara, CA, USA; Figure 4) was used in data analysis and measurement for registration, merging, data cleaning, and smoothing, and saving data in object file (OBJ) format files, including texture information. The saved files were analyzed using Design X (3D Systems, Santa Clara, CA, USA; Figure 5). For accuracy, the measurement takes the breast as separate from the human body. It is assumed that the user area is flat because it could not figure the inside of the body on 3D scanning data.

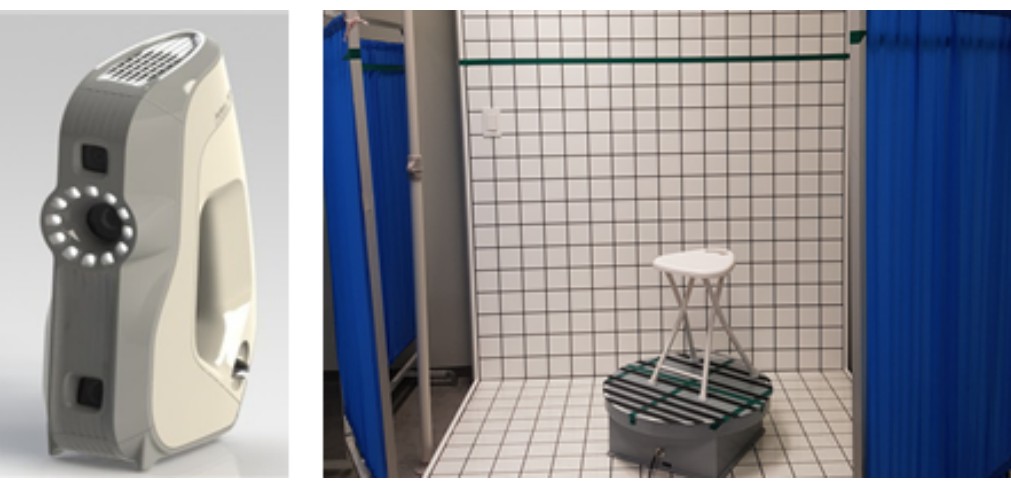

**Figure 4.** Artec Eva 3D Scanner (**left**) Rotating plate for scanning and measurement environment (**right**).

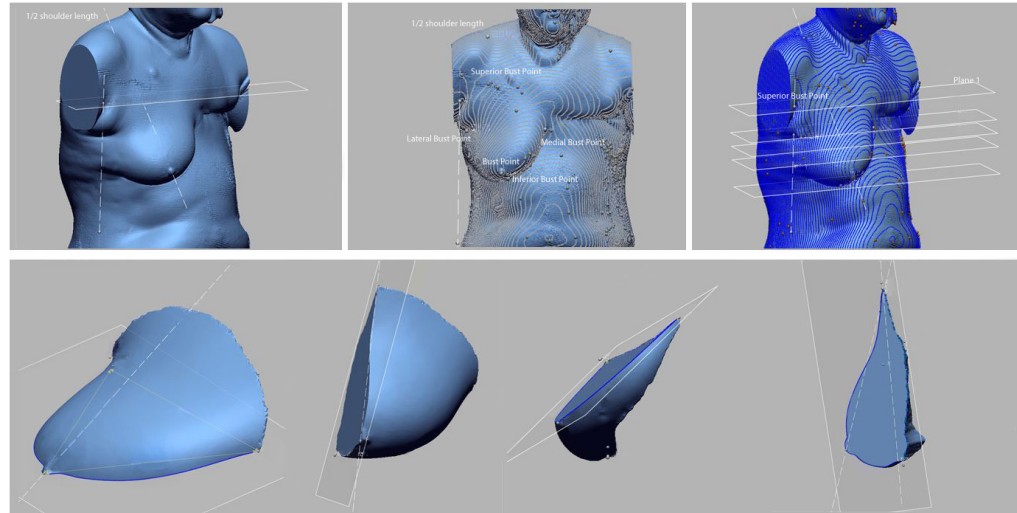

**Figure 5.** Detailed breast item measurement process of 3D-SIM.

### 3.5. Comparison of Breast Volume Data Based on Human Body Data Images

As MRI requires the patient to lie prone on the device and let the breasts droop through the holes of the bed, the image obtained is in a long ellipsoid shape (sloth), whereas 3D-SIM is scanned in a standing position, and thus, the bust rises above the rib cage. Lee et al. [47] conducted extensive research on breasts and assumed them to be in various shapes, such as a cone, arc, and semicircle. Assuming that saggy breasts are cone-shaped, the correlation coefficient was r2 = 0.607, which was suggested as the most suitable method. Eder et al. [48]

claimed that breasts are a complex of various shapes, and thus it is efficient to measure the volume by dividing them rather than applying one cubic form. Thus, this study adopted a method to obtain the volume by dividing breasts by bust point as shown in study [48] rather than measuring a single form, like Lee et al. [47]. As the participants in this study had saggy breasts, this study used the method of obtaining cone volume, which is suitable for saggy breasts, as suggested by Lee et al. [47] (Figure 6). Therefore, using this method, the volume of the breasts in 3D-SIM data was obtained with the assumption that the upper part is in the cone shape and the lower part is in the ellipsoid shape on the sagittal plane. All of the participants' breasts were divided based on the bust point and were separated into upper and lower parts of breasts by connecting the surface line between the medial and lateral bust points around the bust point. The length of the superior and interior bust points was divided into three even parts on average by the connecting surface line between the medial and lateral bust points around the bust point, showing a cone shape up to the point of 2/3, while the bottom part showed the shape of 1/4 ellipsoid, on the sagittal plane. Therefore, to measure the volume of 3D-SIM, the cone volume formula was applied to the upper part of the breast, while the ellipsoid volume formula was applied to the lower part (Formula (1)). This is the 3D-SIM volume calculation formula.

$$3D - SIMV = 1/2 cone volume + 1/4 ellipsoid volume = (b * a^2 * \pi / (3 * 2)) + (a * c * d * \pi * 4 / (3 * 4 * 2)) \quad (1)$$

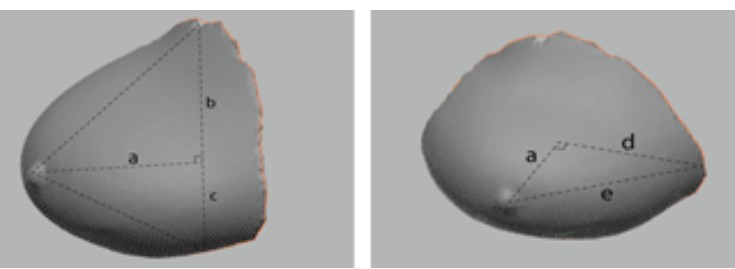

**Figure 6.** The 3D-SIM volume calculation.

For MRI, breasts are in an ellipsoid shape drooping downwards, and thus the volume was measured with the assumption that it is in the ellipsoid shape. MRI images showed the shape of 1/2 ellipsoid sagging downward, and thus the 1/2 ellipsoid volume formula was applied (Formula (2)). This is the MRI volume calculation formula (Figure 7).

The breast volume measured in 3D-SIM and MRI data were analyzed with descriptive statistics and a t-test using IBM SPSS Statistics for Windows Version 22.0 (IBM Corp, Armonk, NY, USA). The formulas are as follows:

$$MRIV = 1/2 ellipsoid volume = (a * b * c * 4) / (3 * 2) \quad (2)$$

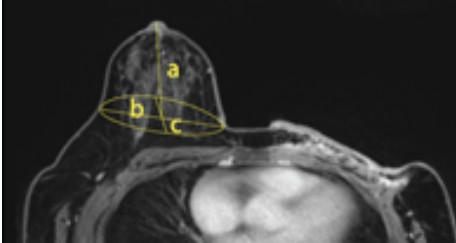

**Figure 7.** The MRI volume calculation.

## 4. Results

### 4.1. Anthropometric Comparison

The average age of the participants in 3D-SIM was 49.75, ranging from 45 to 54. According to the BMI standard, two participants were overweight and two were normal

weight. All four participants had one of their breasts fully removed, owing to breast cancer, did not receive reconstructive surgery, and had breastfeeding experience. They received lymphadenectomy along with breast cancer surgery, but three out of the four patients suffered from edema (Table 5).

**Table 5.** Body measurements of participants.

| Items | A | B | C | D | Mean | S.D. |
|---|---|---|---|---|---|---|
| Age (yr) | 51 | 45 | 54 | 48 | 49.5 | 3.9 |
| Height (cm) | 150 | 157 | 166 | 163 | 159.0 | 7.1 |
| Weight (kg) | 54.6 | 54.7 | 70.4 | 82.9 | 65.7 | 13.7 |
| BMI | 24.3 | 22.2 | 29.6 | 31.2 | 26.8 | 4.3 |
| Total mastectomy | one-side | one-side | one-side | one-side | - | - |
| Lymphedema | yes | no | yes | yes | - | - |
| Bra size | 85B | 90B | 90B | 95B | - | - |

### 4.2. Comparison of Characteristics Based on Human Body Data Images

Among the projective images for medical purposes, anthropometric measurements that can apply anthropometric measurement and the images containing anthropometric data to be used in bra design were MRI data. Therefore, in this study, 3D-SIM and MRI data were chosen for comparison. The results of comparing each character are shown in Table 6.

As image data by postures of 3D-SIM and MRI are different, each item can be different measurements. As follows in Table 6, it was possible to obtain the circumference, depth, breadth, and angle items without a difference in the imaging method. Most items of length can be measured, but MRI could not measure the related shoulder, as most of the MRI focused on the breast and obtained a sectional image. However, for depth, 3D-SIM cannot measure projective items, such as bust point depth, and the breadth or height of the breast base area cannot be measured, owing to the uncertainty of the breast base in 3D-SIM. For height, MRI scans only the upper body, and thus it was impossible to measure the height.

**Table 6.** Comparison of items depending on image acquisition method.

| Measurement Method | Name | 3D-SIM | MRI |
|---|---|---|---|
| Circumference items | Bust surface circumference | yes | yes |
| | Under-bust surface circumference | yes | yes |
| Depth items | Bust depth | yes | yes |
| | Under-bust depth | yes | yes |
| Breadth items | Bust breadth | yes | yes |
| | Under-bust breadth | yes | yes |
| Length items | superior bust-length | yes | yes |
| | Inferior bust-length | yes | yes |
| | Lateral bust-length | yes | yes |
| | Medial bust-length | yes | yes |
| | Superior bust surface length | yes | yes |
| | Inferior bust surface length | yes | yes |
| | Lateral bust surface length | yes | yes |
| | Medial bust surface length | yes | yes |
| | Circumference surface length | yes | yes |
| | Under-bust circumference length | yes | yes |
| | Length between central line and bust point | yes | yes |
| | Shoulder 1/2 length | yes | no |
| | Anterior neck point-bust point length | yes | no |
| Depth items | Chest depth | yes | yes |
| | Bust point depth | no | yes |
| Angle items | Upper bust slope angle | yes | yes |

### 4.3. Anthropometric Comparison

The results of analyzing human body data based on imaging of the participants are shown in Table 7. Ten items showed a statistically significant difference and were found to be affected by the measuring method: bust depth, under-bust depth, bust breadth, under-bust breadth, lateral bust-point length, lateral bust-point surface length, upper bust slope angle, chest depth, bust point height, and circumference surface length.

**Table 7.** Comparison of human body measurement data of participants (unit: cm).

| Items | 3D-SIM | | | | | MRI | | | | | *t*-Value |
|---|---|---|---|---|---|---|---|---|---|---|---|
| | A | B | C | D | Mean(S.D.) | A | B | C | D | Mean(S.D.) | |
| Bust circumference | 94.9 | 93.4 | 94.9 | 109.9 | 98.28 (7.78) | 93.7 | 94.7 | 106.9 | 106.3 | 100.4 (7.17) | 0.41 |
| Under-bust circumference | 89.7 | 88.4 | 87.0 | 103.2 | 92.08 (7.50) | 86.5 | 89.1 | 92.1 | 95.3 | 90.75 (3.80) | −0.31 |
| Bust depth | 21.3 | 21.1 | 20.6 | 27.5 | 22.63 (3.26) | 20.6 | 18.2 | 19.5 | 20.8 | 19.79 (1.19) | 5.50 ** |
| Under-bust depth | 11.1 | 12.6 | 12.0 | 16.4 | 13.03 (2.33) | 20.9 | 17.5 | 17.7 | 19.9 | 18.98 (1.69) | 4.65 *** |
| Bust breadth | 32.1 | 32.4 | 31.0 | 35.7 | 32.80 (2.02) | 31.7 | 29.4 | 32.7 | 35.5 | 32.33 (2.53) | 11.23 *** |
| Under-bust breadth | 30.6 | 31.0 | 34.9 | 32.0 | 32.1 (1.67) | 32.0 | 36.9 | 35.2 | 35.5 | 34.90 (2.07) | 15.42 *** |
| Superior bust slope angle | 29.5 | 26.1 | 27.0 | 24.6 | 26.80 (2.05) | 48.5 | 36.2 | 43.0 | 61.7 | 47.35 (10.8) | 3.74 * |
| Lower bust slope angle | 62.5 | 38.5 | 64.0 | 70.5 | 58.88 (14.0) | 15.5 | 53.9 | 49.0 | 48.3 | 41.68 (17.6) | −1.53 |
| Superior bust length | 12.7 | 12.4 | 13.8 | 12.7 | 12.90 (0.62) | 13.8 | 14.3 | 14.0 | 11.1 | 13.30 (1.48) | 0.49 |
| Inferior bust-length | 5.1 | 5.8 | 6.1 | 5.4 | 5.60 (0.44) | 4.6 | 6.9 | 7.7 | 8.9 | 7.03 (1.81) | 1.51 |
| Lateral bust length | 10.0 | 11.2 | 11.2 | 13.7 | 11.53 (1.56) | 5.2 | 9.9 | 8.4 | 9.6 | 8.28 (2.15) | −2.41 |
| Medial bust length | 13.2 | 14.6 | 13.6 | 13.6 | 13.75 (0.60) | 5.7 | 8.4 | 9.8 | 10.4 | 8.58 (2.09) | −4.76 * |
| Superior bust surface length | 12.6 | 12.4 | 13.8 | 12.8 | 12.90 (0.62) | 14.1 | 14.4 | 14.4 | 11.6 | 13.63 (1.36) | 0.91 |
| Inferior bust surface length | 5.3 | 5.8 | 6.8 | 5.7 | 5.90 (0.64) | 4.6 | 7.0 | 7.7 | 9.8 | 7.28 (2.14) | 1.21 |
| Lateral bust surface length | 9.8 | 11.2 | 11.3 | 14.5 | 11.70 (1.99) | 5.2 | 9.8 | 8.2 | 10.5 | 8.43 (2.36) | −2.14 |
| Medial bust surface length | 13.8 | 14.8 | 14.1 | 14.1 | 14.20 (0.42) | 5.6 | 8.5 | 9.9 | 11.1 | 8.78 (2.37) | −4.52 ** |
| Chest depth | 3.6 | 4.9 | 3.2 | 4.6 | 4.08 (0.81) | 4.5 | 6.5 | 8.4 | 8.8 | 7.05 (1.97) | 2.83 * |
| Bust point depth | 4.3 | 5.4 | 6.5 | 5.1 | 5.33 (0.91) | 6.0 | 6.7 | 9.4 | 9.6 | 7.93 (1.84) | 2.52 * |

\* $p < 0.05$, \*\* $p < 0.01$, \*\*\* $p < 0.001$.

In particular, the difference in items, such as depth and breadth, may be due to the difference between the standing position and prone position, where the depth was lower and the breadth higher on 3D-SIM than on MRI data. The upper bust slope angle on MRI data increased, as the breasts sagging downward were generally affected by gravity by 3D-SIM.

Lateral bust-point length was the item statistically affected by posture among the superior, inferior, medial, and lateral bust-points, surrounding the breasts. In general, breasts spread out in a radial form from the body; thus, the length from bust-point to lateral bust-point was longer than the length from the bust-point to medial bust-point, but as axillary muscles of saggy breasts tend to droop due to lack of elasticity, the lateral bust-point length tends to be longer than the medial bust-point length. Moreover, the lateral bust-point and medial bust-point on 3D-SIM data were longer than MRI data, as the breasts droop downward. In addition, in the case of saggy breasts, the bust circumference line of 3D-SIM data is lower than the bust location of MRI data, owing to gravity. Saggy breasts have lower chest and bust point depth, as they droop downward, compared to hemispherical breasts, but when affected equally by gravity as shown in MRI data, chest and bust point depth were statistically proven to be longer than 3D-SIM.

Lateral bust point length was the item statistically affected by posture among superior bust point, inferior bust point, medial bust point, and lateral bust point surrounding the breasts. In general, breasts spread out in radial form from the body, and thus the length from bust point to lateral bust point was longer than the length from bust point to medial bust point, but since axillary muscles of saggy breasts tend to droop due to lack of elasticity, the lateral bust point length tends to be longer than the medial bust point length. Moreover, the lateral bust point and medial bust point on 3D-SIM data were longer than MRI data since the breasts droop downward. Moreover, in the case of saggy breasts, the bust circumference

line of 3D-SIM data is lower than the bust location of MRI data due to gravity. Saggy breasts have lower chest depth and bust point depth since they droop downward, compared to hemispherical breasts, but when affected equally by gravity as shown in MRI data, chest depth and bust point depth were statistically proved to be longer than 3D-SIM.

*4.4. Comparison of Breast Volume in MRI and 3D-SIM Data*

Breast volume of MRI and 3D-SIM was obtained using the formulas above, and the results of a paired samples t-test to determine whether the two have different data are shown in Table 8. The results were $t = 0.617$ and $p = 0.581$, not showing a statistically significant difference at a significance level of 0.05, indicating no difference in the total volume of the two data (Table 8).

**Table 8.** Paired samples of breast volume data (unit = $cm^3$).

| MRI Volume Data | | 3D-SIM Volume Data | | *t*-Value |
|---|---|---|---|---|
| Mean | S.D.; | Mean | S.D.; | |
| 816.98 | 200.34 | 881.89 | 194.83 | 0.93 |

To validate the size of breast volume obtained from the formulas, this study used the breast volume provided in Hologic R2 and Volpara, which is the mammography system among PPM data of patients (Table 9). Breast volume measured in the program above has a linear relationship of r2 = 0.89 with the actual breast volume and has high predictive power [49], and thus is used as the comparison group to validate the MRI and 3D-SIM volume data. The difference in breast volume between the three measurement methods was statistically analyzed using the Friedman test. The results were $\times 2 = 3.50$ and $p = 0.17$ at a 0.05 significance level, showing no difference between the three volumes. This experiment proved that the measurement of breast volume is statistically similar regardless of breast shape.

**Table 9.** Paired samples t-test of breast volume data (unit = $cm^3$).

| MRI Volume Data | | 3D-SIM Volume Data | | Mammography Data | | *x*2(*p*) |
|---|---|---|---|---|---|---|
| Mean | S.D.; | Mean | S.D.; | Mean | S.D.; | |
| 816.98 | 200.34 | 881.89 | 194.83 | 811.53 | 132.03 | 3.50 (0.17) |

## 5. Discussion

A total of 68.4% of breast cancer patients worldwide choose mastectomy for breast cancer treatment. It is because mastectomy has been effective in preventing active cancer metastasis so far [50,51]. Mastectomy patients have been wearing mastectomy bras or breast prostheses to supplement their physical body and for body image treatment [52,53]. Patients who choose mastectomy for breast cancer treatment suffer not only psychological pain due to body loss, but also physical pain due to body imbalance. Therefore, it is recommended to wear a bra with an artificial breast for excision patients. However, as wearing a bra and artificial breast that do not fit the body well increases pain, it is necessary to manufacture an individually tailored bra and artificial breast.

Body data are obtained through body scanning using anthropometric methods that can be used with 3D-SIM data to design customized mastectomy bras. However, there is the discomfort of having to reveal one's body to others. In addition, the anthropometric posture causes the breasts to sag, thereby not perfectly revealing the breast characteristics.

Thus, this study was designed to test the hypothesis that, as the breast consists of adipose tissue and a few muscles, breasts have consistent volume regardless of shape, owing to posture. Additionally, the relevant volume measurements from the medical image, not the anthropometric image data, can be used to make mastectomy bras. Therefore, this

study used image data for medical purposes as human body data that can limit exposure to the human body while well-displaying breast characteristics. MRI can scan the inside of the human body, skeletal structure, and outline simultaneously, using nuclear magnetic resonance, while 3D-SIM can film the surface of the human body, using white structured light. Both methods are contactless measurements using energy sources that are harmless to the human body. As for posture, MRI requires the patient to lie prone on the device and let the breasts droop through the holes of the bed; therefore, the bust will be in an ellipsoid shape (sloth) and is completely separated from the body.

However, 3D-SIM requires the patient to be in a standing position; thus, the breasts are affected by gravity. Saggy breasts or big breasts tend to hang down, which folds with the body, making it difficult to accurately measure the under-bust circumference. Moreover, saggy breasts have a bust circumference line lower than the under-bust circumference line, which makes it difficult to define the accurate position of the anthropometric bust circumference line. MRI measures parts that are not visibly identifiable, such as the rib cage and breast base area, but the images are sectional, and thus additional processing is required to obtain the overall human body shape data. Moreover, the measurement points cannot be marked, owing to the sectional composition, and the scope of image acquisition is not consistent depending on the technician and purpose, thereby omitting parts that can be the standard for body measurements, such as anterior neck point, shoulder, and back. While 3D-SIM can obtain 3D data of the body surface and thus can apply the same measurement method as anthropometry, the projection of the breast base form and under-bust outline is unclear, thereby possibly distorting the breast base area when the breasts are too saggy. In addition, as the under-bust circumference line cannot be identified, the participants may be requested to rescan the under-bust circumference line by correcting the saggy breasts, thereby increasing their fatigue.

As PPM by MRI and 3D-SIM are scanned in different postures, the shape of the scanned breasts is also different. The measurements of the detailed items of breasts were also different in the two forms. By comparing the two data, 18 items with matching items were measured. MRI data for medical purposes and anthropometric data of 3D-SIM showed statistically significant measurements differences in 10 items: bust depth, under-bust depth, bust breadth, under-bust breadth, lateral bust point length, lateral bust point surface length, upper bust slope angle, chest depth, bust point height, and circumference surface length. Moreover, in 3D-SIM, the breasts overlap with the upper body for the under-bust circumference line and inferior bust point. Thus, the under-bust circumference line was secured by overlapping with the image securing the under-bust circumference line with different arm angles. The items showing differences in posture were breast thickness, breast width, and lower breast width. Compared with the 3D data, the PPM data showed a decrease in thickness and an increase in width. In addition, in the data of 3D-SIM, the breast is sagging. In addition, the average age of the subjects was 49.75 years; the previous research result, which proved that breast sagging occurred due to age in the 40s and older, was supported [54]. Therefore, 3D-SIM data showed that the length of the lateral point was longer than MRI data, and the upper breast inclination angle was smaller. Although no statistically significant difference could be found, by taking the MRI measuring posture, a tendency for the length between the superior bust point to the inferior bust point tending to be longer than the 3D-SIM data was shown, but the length between the medial and lateral tended to be shorter. In general, the length between the lateral bust point and bust point is longer than the length between the medial and bust point; however, in the case of PPM, there is no such difference, suggesting the possibility of eliminating the variable of breast deformity due to sewage of the breast. In addition, it was statistically proven that the depth of the chest and the depth of the bust point greatly increased in the PPM data [55]. Moreover, to measure the breast volume, the breast of 3D-SIM data was separated into two parts by a line between the lateral and the medial bust points through the bust point. For the upper part, the formula for calculating the cone volume was applied, while the hemispherical volume calculation formula was applied to the lower part. Therefore,

breast volume for PPM by MRI and 3D-SIM was measured to identify the applicability of MRI data, which proved that there was no statistical difference in volume. In other words, making bras using MRI data with high human body reproducibility can reduce additional exposure or contact with the human body, thereby seeking the mental stability of mastectomy patients and applying all kinds of data collected in the treatment process to bra design. Moreover, by applying the characteristics of non-resected breasts to the design, the prototype bras can be adequately used on non-resected breasts as well. Using MRI data can make up for the insufficient human body data of total mastectomy patients in Korea, and the diversity of excision can be reflected in the bra design. Therefore, by using MRI data, custom bras can be made without additional body measurements.

Despite the strength of this study, there remain two major limitations. First, in the case of MRI data, although the breast is completely separated from the rib cage and the volume of the breast can be appropriately measured, there is a disadvantage that the projection range is limited to the torso. Moreover, the method and scope of taking pictures are not standardized, compared to taking images for the 3D-SIM. As MRI data scan only the torso of the human body, the scope of scanning may vary depending on the technician, thereby cropping some parts occasionally. Therefore, it is necessary to standardize the scope of scanning to use MRI data for such purposes as making bras. The second major limitation of this study is that only four mastectomy patients agreed to participate. However, the significance of the study lies in the fact that it has provided the possibility of producing bras and breast prostheses for mastectomy patients using medical images. In the future, we plan to design a drawing method that can produce a bra by setting a formula suitable for bra production using the changed length and conducting an experiment to verify the fit. Additionally, further research can be conducted on making bras based on the results of this study for a healthy body and improved clothing habits for mastectomy patients.

**Funding:** This research was funded by the Korean Breast Cancer Society (350-20170074) and https://www.kbcf.or.kr/index.do (accessed on 20 May 2018).

**Institutional Review Board Statement:** The study was conducted according to the guidelines of the Declaration of Helsinki, and approved by Institutional Review Board (IRB) of Seoul National University Hospital (IRB No. 1007-211-325) and the Ethics Committee of Seoul National University Research Ethics Team (IRB No. 1808/003-001).

**Informed Consent Statement:** Informed consent was obtained from all subjects involved in the study.

**Conflicts of Interest:** The author declared no potential conflicts of interest with respect to the research, authorship, and/or publication of this article.

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
