# Peer review of "Verification of Usability of Medical Image Data Using Projective Photography for Designing Clothing for Breast Cancer Patients"

_tomography, doi:10.3390/tomography8040153_

Round 1
Reviewer 1 Report
- The main question addressed by the research is to verify if Projective photography (a medical image data method) can be used as a valid method for clothing pattern making (for mastectomy bras).
- The topic is relevant in its subject area because anthropometric methods of measure cause mistakes because of the body posture is different.
- This manuscript adds to the field a new validation of other method to measure breast size in order to make customized mastectomy bras.
- I consider that metodology used is appropriate for the study carried out.
- In this manuscript there is no section explicitly called Conclusions, but in Discussion author address the main question of the study and presents consistent arguments to validate the results of the study.
- References are appropriate and well presented, despite not including all the authors in some of them (7, 12, 16, 31, 42 and 43).
- Tables and figures also are well presented and help to understand the methods and the results.
-
The manuscript entitled `A study on the verification of medical image data (Projective photography) usability for clothing Pattern making´ and presented by Youn Joo Kim is an interesting study about the use of medical images as an alternative to anthropometric methods (in wich the shape of the breast is usually deformed because the different posture between both methods) to manufacture custumized mastectomy bras in order to retrieve the body balance of the patients and to reduce their psychological suffering. Author explains that if the volumen is constant, mastectomy bras could be manufactured using any of them. When the volume is calculated by separating the shape of the breast into a cone and an hemispherial shape (3D-SIM) of applying an ellipsoidal volume formula (MRI) there was no significant differences using a t -test, proving that the volume of the breast does not change. Author should present the `non-published material´ in english.
Author Response
[5. July, 2022]
Dear Editor:
I wish to submit a research paper for publication in Tomography, titled “A study on the verification of medical image data (projective photography) usability for clothing pattern making in breast cancer patients.
This study aimed to verify the usability of image data for medical purposes in designing clothing for breast cancer patients. In using anthropometric method, the resulting medical image of the breast is deformed, making it difficult to obtain an accurate visual measurement for drafting a mastectomy bra pattern. This study tested the hypothesis that breast volume is constant despite the difference in measurement methods using 3D-SIM and PPM by MRI. I believe that our study makes a significant contribution to the literature because it focuses on functionality, and the results can help mastectomy patients improve their mobility, self-image, and self-confidence.
Further, I believe that this paper will be of interest to the readership of your journal because it tests the usability of medical imaging in designing clothing for breast cancer patients.
Moreover, I am so glad to get the specific comment. I have got another chance to revise my article once again.
I tried to figure out what is not suited for this article(also, several errors)
I asked an expert to correct my English to make better sentences in English.
And I modified the references to mention all of the authors.
This manuscript has not been published or presented elsewhere in part or in entirety and is not under consideration by another journal. All study participants provided informed consent, and the study design was approved by the appropriate ethics review board. I have read and understood your journal’s policies, and we believe that neither the manuscript nor the study violates any of these. There are no conflicts of interest to declare.
Thank you for your consideration. I look forward to hearing from you.
Sincerely,
Kim, Youn Joo.
Ph D. in Seoul National University
teru281@gmail.com

Reviewer 2 Report
Dear Author,
I congratulate you on the very interesting topic. Your results on the evaluation of MRI data to manufacture a customized bra for mastectomized women could be a stimulus towards a more in-depth evaluation of the argument and further studies. Indeed, the major limitation of your study is the small sample. Nevertheless, as you point out in the conclusions, I stimulate you on proceeding with the planned further study.
Furthermore, an extensive review of the English language is mandatory. There are many typos and syntax errors, some sentences are unclear and not well written or easily understandable (see for example, lines 1-7 of the abstract; lines 31-33; lines 55-57; etc.).
Given these consideration, I think that your paper could be accepted for publication after these other corrections:
1. Since you evaluate the manufacture of a customized bra for breast cancer patients, please, considered to add this aspect (“…in breast cancer patients”) in the title.
2. Introduction: according also to the suggested review of the English language, use <%> instead of percentage (lines 21-28); furthermore the sentence “that is surgical…tissue” on line 17 is unnecessary; line 47: (CT) instead of CT.
3. Table 1 is not clear since there is no separation between rows in the “characteristic section” between the different techniques. The same happens in Table 2 and 3.
4. In table 2, 3, 4 and 6, please, explain the abbreviations (e.g. L.; D.; C.; BP; etc.)
5. Also in the text there is no explanation for BP; please, consider to add.
6. Material and Methods: is it S University Hospital or Seoul University Hospital?
7. Figure 1. I think that C represent the Lateral Bust Point and D the medial bust point or consider to modify the letters in the figure.
8. Lines 250-251 are repetition of lines 248-249; please, erase them.
9. Results: line 257, did you mean lymphadenectomy instead of lymphoidectomy?
10. Results: paragraph 4.2. Lines 264-287 are very interesting but I think they fit better in the discussion section. Please, consider to move these lines. Indeed, I think that in this paragraph, you should add only the results, summarizing and highlighting the differences between MRI and 3D-SIM in according to table 6.
11. Results: Line 275 is redundant (“can scan both inside and outside…thus”), consider to write directly MRI can measure, etc.
12. Results: lines 288-291. To what they refer?
13. Discussion: line 339; why patients chose mastectomy? Are you sure of this statement?
14. Discussion: line 367 and 376; the statistical significance is related to an evaluation done in your paper or are they references?
Kind Regards
Author Response
[5. July, 2022]
Dear Editor:
I wish to submit a research paper for publication in Tomography, titled “A study on the verification of medical image data (projective photography) usability for clothing pattern making in breast cancer patients.
This study aimed to verify the usability of image data for medical purposes in designing clothing for breast cancer patients. In using anthropometric method, the resulting medical image of the breast is deformed, making it difficult to obtain an accurate visual measurement for drafting a mastectomy bra pattern. This study tested the hypothesis that breast volume is constant despite the difference in measurement methods using 3D-SIM and PPM by MRI. I believe that our study makes a significant contribution to the literature because it focuses on functionality, and the results can help mastectomy patients improve their mobility, self-image, and self-confidence.
Further, I believe that this paper will be of interest to the readership of your journal because it tests the usability of medical imaging in designing clothing for breast cancer patients.
Moreover, I am so glad to get the specific comment. I have got another chance to revise my article once again.
I tried to figure out what is not suited for this article(also, several errors) as follows your comments and revise that.
First of all, I tried to make a clear sentence what you advised.
I asked an expert to correct my English.
From now on, I would mention as your comment numbers what I revised.
1. I mentioned “breast cancer Patient” in the title and changed the whole order more soften it.
2. I had no idea how I wrote % mark, however, I figured it out finally. I changed all. And I deleted the unnecessary sentences. And I removed the parenthetical definitions for CT and MRI by my English adviser.
3. I modified the tables to make them easier to recognize.
4. 5. I decided not to use abbreviations. All the abbreviations have been corrected.
6. I changed to Seoul National University Hospital.
7. I modified figure 1 and table 3.
8. I deleted repetition sentences.
9. I changed the word to lymphoidectomy
10. Some of the sentences were moved to discussion sessions and I focused to describe the differences.
11. I revised the sentences in the Result parts
12. I modified the errors
13. I mentioned the reason why patients choose mastectomy by reference.
14. I added references to support my opinion.
This manuscript has not been published or presented elsewhere in part or in entirety and is not under consideration by another journal. All study participants provided informed consent, and the study design was approved by the appropriate ethics review board. I have read and understood your journal’s policies, and we believe that neither the manuscript nor the study violates any of these. There are no conflicts of interest to declare.
Thank you for your consideration. I look forward to hearing from you.
Sincerely,
Kim, Youn Joo.
Ph D. in Seoul National University
teru281@gmail.com

Reviewer 3 Report
The article presents a verification of the usefulness of medical image data for creating clothing patterns.
The paper is an interesting approach in the aspect of using medical images. Data from medical images are obtained repeatedly in the treatment process and thus can analyze various changes in the human body before and after treatment. Nevertheless, in imaging-based methods, the scope and quality of the data is an important element. Effectiveness depends largely on the selection of criteria and the amount of data and the appropriate selection of data. Universality thus has some limitations.
Minor remarks:
1) I suggest you describe in more detail on what basis and from what the scope of data selection for the research problem posed.
2) What is the scientific essence of the article, originality and main contribution to the presented scientific field?
3) The article needs minor language corrections.
Author Response
[5. July, 2022]
Dear Editor:
I wish to submit a research paper for publication in Tomography, titled “A study on the verification of medical image data (projective photography) usability for clothing pattern making in breast cancer patients.
This study aimed to verify the usability of image data for medical purposes in designing clothing for breast cancer patients. In using anthropometric method, the resulting medical image of the breast is deformed, making it difficult to obtain an accurate visual measurement for drafting a mastectomy bra pattern. This study tested the hypothesis that breast volume is constant despite the difference in measurement methods using 3D-SIM and PPM by MRI. I believe that our study makes a significant contribution to the literature because it focuses on functionality, and the results can help mastectomy patients improve their mobility, self-image, and self-confidence.
Further, I believe that this paper will be of interest to the readership of your journal because it tests the usability of medical imaging in designing clothing for breast cancer patients.
Moreover, I am so glad to get the specific comment. I have got another chance to revise my article once again. Also, I revised to background and references what supported to my argument. Moreover I modified the reference errors.
From now on, I would mention as your comment numbers what I revised.
- I revised how I selected the subject by criteria in the article
- The academic significance of this paper is as follows; firstly, it suggested the possibility of converging the categories of scientific domains while it discovered that projective image data is used not only for medical purposes but also for Clothing Ergonomics purposes.
Second, it showed the possibility of changing the perception that measurements by the unstandardized posture that is not followed as anthropometric manual was not proper.
- I tried to figure out what is not suited for this article(also, several errors) and revised. I asked an expert to correct my English to make better sentences in English.
This manuscript has not been published or presented elsewhere in part or in entirety and is not under consideration by another journal. All study participants provided informed consent, and the study design was approved by the appropriate ethics review board. I have read and understood your journal’s policies, and we believe that neither the manuscript nor the study violates any of these. There are no conflicts of interest to declare.
Thank you for your consideration. I look forward to hearing from you.
Sincerely,
Kim, Youn Joo.
Ph D. in Seoul National University
teru281@gmail.com
